# Glial Fibrillary Acidic Protein’s Usefulness as an Astrocyte Biomarker Using the Fully Automated LUMIPULSE^®^ System

**DOI:** 10.3390/diagnostics14222520

**Published:** 2024-11-11

**Authors:** Jo Kamada, Tomohiro Hamanaka, Aya Oshimo, Hideo Sato, Tomonori Nishii, Marika Fujita, Yoshiharu Makiguchi, Miki Tanaka, Katsumi Aoyagi, Hisashi Nojima

**Affiliations:** FUJIREBIO Inc., 1-8-1 Akasaka, Minato-ku, Tokyo 107-0052, Japan; jo.kamada@hugp.com (J.K.); tomohiro.hamanaka@hugp.com (T.H.); aya.oshimo@hugp.com (A.O.); hideo.sato@hugp.com (H.S.); tomonori.nishii@hugp.com (T.N.); marika.fujita@hugp.com (M.F.); yoshiharu.makiguchi@hugp.com (Y.M.); miki.tanaka@hugp.com (M.T.); katsumi.aoyagi@hugp.com (K.A.)

**Keywords:** glial fibrillary acidic protein, neurological biomarker, chemiluminescent enzyme immunoassay

## Abstract

Background: Glial fibrillary acidic protein (GFAP) is an important biomarker for neuroinflammatory conditions. Recently, advancements in the treatment of neurological diseases have highlighted the increasing importance of biomarkers, creating a demand for accurate and simple measurement systems for GFAP levels, which are essential for both research and clinical applications. This study presents the development and validation of a novel fully automated immunoassay for the quantitative determination of GFAP levels in biological samples. Methods: We examined the analytical performance of the GFAP assay on the LUMIPULSE platform. The assay’s parameters, including antibody concentrations, incubation times, and detection methods, were optimized to enhance sensitivity and specificity. GFAP levels were measured in 396 serum or plasma samples, comprising both healthy controls and patients with neurodegenerative diseases. Results: In the analytical performance studies, intra- and inter-assay coefficients of variation (CV) were below 5%, indicating high reproducibility. Additionally, the assay demonstrated good linearity over the measurement range. The limit of quantification (LoQ) for this assay was 6.0 pg/mL, which is sufficient for measuring specimens from healthy controls. In clinical validation studies, GFAP levels were significantly elevated in patients with neurodegenerative diseases compared to healthy controls. Conclusions: This automated GFAP assay provides a robust and reliable tool for GFAP measurement, facilitating further research into GFAP’s role in neurological disorders and potentially aiding in the diagnosis and monitoring of these conditions.

## 1. Introduction

The central nervous system (CNS) is a complex and vital system responsible for processing and transmitting information throughout the body [1]. Comprised of the brain and spinal cord, the CNS is supported by various cell types, including neurons and glial cells [1,2]. Neurons, the primary functional units of the nervous system, transmit signals through synapses, facilitating communication between different brain regions and between the brain and peripheral organs. Meanwhile, glial cells, particularly astrocytes, provide critical support and maintenance functions [1,2]. Among the various proteins expressed by astrocytes, glial fibrillary acidic protein (GFAP) serves as a key biomarker for astrocytic activity and CNS integrity [3,4].

GFAP is an intermediate filament protein specifically expressed in astrocytes, which comprise approximately 20% of the cells in the human brain [2,5]. These cells perform essential functions in the CNS, including maintaining the blood–brain barrier, regulating neurotransmitter levels, and responding to injury [2,6]. In pathological conditions, such as traumatic brain injury (TBI), neurodegenerative diseases, and inflammatory disorders, astrocytes become activated and increase their GFAP expression [7,8]. As a result, GFAP levels in the blood can reflect astrocytic activity and provide valuable insights into underlying pathological processes [4]. Elevated GFAP levels in the blood have been associated with various neurological disorders, including Alzheimer’s disease (AD), multiple sclerosis (MS), and TBI [4,7,8,9,10]. This indicates that GFAP is not only a marker of astrocytic activation, but also serves as a potential prognostic indicator in acute neurological conditions, enabling clinicians to assess patient prognosis and tailor treatment strategies accordingly [4,10].

Traditional methods for measuring GFAP, such as enzyme-linked immunosorbent assays (ELISAs), are commercially available [11]. However, an ELISA is often time-consuming and requires a high level of technical proficiency, which can adversely affect accuracy and reproducibility. Moreover, detecting GFAP in blood has historically been challenging due to limited sensitivity, particularly for low GFAP concentrations [4]. New highly sensitive assays, such as single-molecule arrays (SIMOAs), have been developed to measure GFAP levels in blood from healthy individuals and those with various neurological diseases [12]. Although current commercial SIMOAs feature a semi-automated procedure, a considerable degree of technical expertise is still necessary to ensure optimal results. These limitations hinder timely diagnosis and monitoring of neurological conditions, which is critical in acute care settings.

To address the limitations of current GFAP testing methods, a fully automated blood measurement kit for GFAP is required. A combination kit for GFAP and ubiquitin C-terminal hydrolase 1 (UCH-L1) has already received FDA approval for diagnosing TBI [13]. However, there is a demand to investigate GFAP concentrations in other neurological disorders. Furthermore, that combination kit is not approved as an IVD tool in Japan, and not widely distributed there. Therefore, we developed a fully automated blood measurement kit for GFAP using the LUMIPULSE platform. This kit was designed to improve the accessibility and efficiency of GFAP testing in clinical laboratories, providing rapid and reliable results. By automating the entire testing process, we aimed to increase throughput, reduce human error, and ensure consistent results. Our GFAP measurement kit utilizes highly specific antibodies and a streamlined protocol to deliver accurate and reproducible measurements. Furthermore, the LUMIPULSE platform offers a comprehensive portfolio of AD-related biomarkers, facilitating sequential measurements and comprehensive analyses of neurological conditions [14,15,16,17]. In this paper, we present the results of our evaluation of the assay’s basic performance and its clinical utility in testing blood samples.

## 2. Materials and Methods

### 2.1. Development of Mouse Monoclonal Antibodies Against GFAP

Mouse monoclonal antibodies against GFAP were developed to enable specific detection and quantification of GFAP in various biological samples. These antibodies were generated using standard hybridoma technology [18], in which spleen cells from mice immunized with recombinant full-length GFAP were fused with myeloma cells. Following fusion, hybridoma cells were screened for their ability to produce antibodies that specifically recognize GFAP. Positive clones were subsequently expanded, and the antibodies were purified for further characterization. Antibody pairs targeting the core region of GFAP were then screened and applied in immunoassays.

### 2.2. Description of the Fully Automated Immunoassay for GFAP

The GFAP assay was developed on a fully automated chemiluminescent enzyme immunoassay system, LUMIPULSE G1200 and LUMIPULSE G600II (FUJIREBIO Inc., Tokyo, Japan). Briefly, magnetic beads were coated with a monoclonal anti-GFAP mouse antibody that recognizes the core region. GFAP molecules present in specimens are captured by these magnetic beads, generating stable immune complexes. The magnetic beads are then washed to eliminate unbound material and incubated with another monoclonal anti-GFAP mouse antibody conjugated with alkaline phosphatase. After a second wash, 3-(2′-spiroadamantyl)-4-methoxy-4-(3″-phosphoryloxy)-phenyl-1,2-dioxetane (AMPPD) substrate is added to the reaction mixture and the resulting luminescence is measured at 477 nm. The intensity of the luminescent reaction is directly proportional to the concentration of GFAP in the test sample. The sample volume needed for the assay is 100 μL (plus 100 μL of dead volume on the analyzer) and results are available in 30 min (Figure 1). The assay utilizes the full-length recombinant human GFAP (rhGFAP) protein (LS Bio, Shirley, MA, USA) as a reference standard. The reportable range of the assay is 4.0 to 5000.0 pg/mL.

### 2.3. Characterization of In-House Standard Material and Concentration Assignment

rhGFAP was utilized as the in-house standard material, as there are currently no international standard materials for GFAP in vitro diagnostic reagents. The concentration of rhGFAP was determined through amino acid composition analysis (Creative Proteomics, Shirley, NY, USA). GFAP measurement values for the GFAP assay were assigned based on assay calibrators prepared from rhGFAP.

### 2.4. Evaluation of Analytical Performance of the GFAP Assay

#### 2.4.1. Limit of Detection and Limit of Quantitation

The limit of detection (LOD) and limit of quantitation (LOQ) for the GFAP assay were generated, validated, and implemented in the reporting system according to the guidelines outlined in CLSI EP17-A2. The LOQ was determined at an acceptable examination imprecision of CV ≤ 10%.

#### 2.4.2. Precision Studies

Precision was evaluated according to CLSI EP05-A3 using duplicate assays in two analytical runs per day of 20 measurement days within 33 days, with a new aliquot of 3 levels of frozen samples each day. Results were reported as the CV.

#### 2.4.3. Dilution Linearity

Dilution linearity studies were conducted using high-titer plasma or serum specimens spiked with rhGFAP. A dilution series (1×, 2×, 5×, 10×, and 16×) of samples 1 to 10 was prepared using LUMIPULSE *G* Specimen Diluent 1 (FUJIREBIO Inc.). Regression analysis of the observed diluted concentrations was compared to the expected values based on the corresponding concentrations of the undiluted specimen.

#### 2.4.4. Interfering Substances and Cross Reactant

The influence of interfering substances was verified using Interferences Check A Plus (SYSMEX CORPORATION, Kobe, Japan), Intralipid 20% emulsion (Sigma-Aldrich CO. LLC, St. Louis, MO, USA), and Biotin (Tokyo Chemical Industry Co., Ltd., Tokyo, Japan). Three levels of plasma or serum samples were mixed with free bilirubin, conjugated bilirubin, chyle, hemoglobin, triglycerides, and biotin. The mean GFAP value of each test sample was compared with the mean value of the corresponding control sample.

The cross-reactivity toward intermediate filament proteins similar to GFAP was verified using recombinant human desmin (rhDesmin; ProSpec-Tany TechnoGene Ltd., Rehovot, Israel), recombinant human neurofilament light chain (rhNfL; CUSABIO TECHNOLOGY LLC, Houston, TX, USA), recombinant human peripherin (rhPeripherin; Biomatik, Cambridge, ON, Canada), and recombinant human vimentin (rhVimentin; Abcam, Cambridge, UK). Two levels of plasma or serum samples were mixed with rhDesmin, rhNfL, rhPeripherin, and rhVimentin. Mean GFAP values for the control and test samples were determined. The cross-reactivity of each cross-reacting substance was calculated as follows: cross-reactivity (%) = [(mean of test sample) − (mean of control sample)]/(concentration of cross-reacting substance).

#### 2.4.5. Serum and Plasma Correlation

Passing–Bablok regression analysis was conducted to evaluate the correlation be-tween serum and plasma measurement values. Paired samples, either a normal matched set purchased from Precision for Medicine (Frederick, MD, USA) or paired specimens spiked with rhGFAP, were used. Serum and plasma samples were analyzed using the GFAP assay.

### 2.5. GFAP Assay in HC, AD, BT, and CI Samples

GFAP was measured in 396 serum or plasma samples, including 296 from healthy control (HC) individuals, 21 from brain tumor (BT) patients, 10 from cerebral infarction (CI) patients, and 69 from Alzheimer’s disease (AD) patients. Samples were sourced from Japanese volunteers associated with our company, who provided informed consent prior to enrollment, as well as biobank samples from the Tsukuba Medical Laboratory of Education and Research (TMER, Tsukuba, Japan) and commercially available specimens from Discovery Life Sciences (DLS, Huntsville, AL, USA).

### 2.6. Statistics

Precision data analysis was conducted using Analyse-it for Microsoft Excel (info@analyse-it.com; http://www.analyse-it.com (accessed on 18 April 2024); Microsoft, Seattle, WA, USA), while all other statistical analyses were performed with R version 4.4.1 (R Foundation for Statistical Computing, Vienna, Austria; URL: http://www.R-project.org/, (accessed on 10 July 2024)). Continuous variables were summarized as median, first quartile (Q1), and third quartile (Q3). Group comparisons for continuous variables were performed using the Wilcoxon rank-sum test with Bonferroni correction. Pearson correlation coefficients (r) or Spearman correlation coefficients (ρ) were calculated. The significance level for statistical analyses was set at *p* < 0.05.

## 3. Results

### 3.1. Evaluation of Analytical Performance of GFAP Assays

We developed the GFAP assays and evaluated the basic performance of these assay kits. LOD and LOQ were 1.8 pg/mL and 6.0 pg/mL, respectively (Table 1). Precision CVs were as follows: repeatability, 1.8~2.5%; between run, 1.4~1.7%; between day, 2.3~3.4%; and within laboratory 3.5~4.4% (Table 2). The precisions of the kits were less than 5% CV, indicating that the GFAP assay produced favorable results.

Dilution linearity was assessed using serial dilutions (from 1:2 to 1:16) of ten samples. Equations obtained by linear regression showed an R-squared of 1.00 (Table 3).

The influence of interfering substances is shown in Table 4. Mean percentage differences between test samples and control samples ranged from −2.7% to 2.4%.

Cross-reactivity toward intermediate filament proteins similar to GFAP was confirmed by results shown in Table 5. Mean percentage differences between test samples and control samples ranged from −0.6% to 0.2%.

In the correlation test between serum and plasma, results for 153 paired samples yielded a regression equation of *y* = 1.0*x* + 0.6, and the correlation coefficient (*r*) was 1.0, indicating excellent results for both the slope and the correlation coefficient (Figure 2).

### 3.2. Comparison of GFAP Assay Across Different Neural Disease Groups

We evaluated the clinical performance of the GFAP assay using commercially available samples. The age and gender distributions of the subjects are shown in Table 6. GFAP levels in analyzed samples ranged from 10.0 to 1089.9 pg/mL (Figure 3). The median [Q1, Q3] GFAP values of HC, AD, BT, and CI subjects were 23.1 [19.1, 30.0], 100.5 [65.1, 140.6], 63.4 [45.1, 153.9], and 50.4 [38.6, 109.4], respectively. Compared to the HC group, there were statistically significant differences (*p* < 0.001) in AD, BT, and CI subject groups (Figure 3). We further analyzed GFAP levels in the HC group in relation to age and found that GFAP levels exhibited a weak positive correlation with age (Figure 4).

## 4. Discussion

In this paper, we present a high-throughput and highly specific GFAP assay developed using the LUMIPULSE system. To enhance sensitivity and throughput, we employed monoclonal antibodies targeting the core region of GFAP. The assay is fully automated and can complete measurements within 30 min, with a maximum throughput of 120 tests per hour.

Recently, GFAP and UCH-L1 utilized in a handheld device and a fully automated system have received FDA approval, establishing their role as biomarkers for the timely and accurate assessment of TBI [13,19]. Although both platforms demonstrate a strong correlation in their results, the fully automated system tends to yield lower GFAP measurement values [19]. This discrepancy raises concerns regarding the validity of the measured values. Standardization of measurement is crucial for improving healthcare, as it ensures consistent results across different clinical settings and time points. This consistency is essential for the effective application of evidence-based medicine and the development of standardized diagnostic and therapeutic guidelines. To achieve standardization, it is necessary to ensure that measurement values are aligned from the highest hierarchical level down to routine clinical laboratory methods. One key element of this standardization effort is the establishment of international reference standard materials [20]. Currently, GFAP measurements lack such reference materials. To address this gap, we utilized highly purified rhGFAP, with its concentration determined through amino acid component analysis, as an in-house standard. This approach enabled us to trace the obtained GFAP values back to this reference material, thereby enhancing the reliability of our measurements.

We rigorously evaluated the analytical performance of our GFAP assay. The assay demonstrated excellent precision and linearity within the measuring range. Previous studies have reported GFAP concentrations in patients with AD to be generally below 800 pg/mL as measured by SIMOA assay [21]. Our assay can accurately measure GFAP levels up to approximately 5000 pg/mL, covering the range relevant to AD patients. Furthermore, we confirmed the assay’s resistance to interference from endogenous substances, high biotin concentrations, and other intermediate filament proteins.

Using this assay, we successfully quantified GFAP levels in plasma or serum samples from both healthy controls and patients with AD, BT, and CI. Importantly, all healthy control samples exhibited GFAP levels above the functional sensitivity threshold, indicating the assay’s ability to accurately measure low GFAP levels in healthy individuals (Figure 3).

Previous studies have demonstrated a positive correlation between age and GFAP expression in healthy individuals [4,22]. Therefore, we explored the correlation between blood GFAP levels and age and revealed a weak correlation (ρ = 0.230), with a subset of individuals over 50 years of age exhibiting low GFAP levels (Figure 4). One potential explanation for this discrepancy is that individuals over 50 years of age may have amyloid burdens in their brains, which could contribute to increased levels of GFAP. Notably, approximately 17 percent of cognitively normal individuals aged 50 to 54 showed evidence of amyloid abnormalities based on cerebrospinal fluid (CSF) tests, with prevalence increasing with age [23]. To further explore this relationship, future studies should include detailed clinical information, particularly regarding amyloid accumulation. Additionally, recent studies have indicated a significant association between GFAP levels and diabetes, demonstrating that GFAP levels are elevated in diabetic animals, particularly in relation to neuroinflammatory processes [24,25]. The inflammatory response mediated by astrocytes in the diabetic brain may contribute to cognitive decline [26], highlighting the importance of GFAP in understanding the neuroinflammatory landscape of diabetes. Further investigation is needed to elucidate the underlying mechanisms of these associations.

This study had several limitations. First, the lack of comprehensive analyses using specimen information, a small study population, and the absence of a validation cohort limit the generalizability of the findings. Second, the volunteer population consisted of healthy individuals aged 20–50, potentially excluding the effects of underlying diseases such as diabetes mellitus. Future studies should investigate patient specimens across a range of medical conditions within a multidisciplinary hospital setting to establish cutoff points. Third, this study was conducted exclusively on Japanese volunteers, necessitating comparisons with other racial groups. Forth, the unavailability of an FDA-approved GFAP kit in Japan precluded direct correlation testing between our assay and the FDA-approved kit. Fifth, although we reported the measurement of serum and plasma samples in this study, we acknowledge that the measurement range for CSF samples needs to be expanded, which will be addressed in future research. Finally, further research is needed to establish definitive cutoff values for GFAP levels.

## 5. Conclusions

In conclusion, this study successfully demonstrated that the GFAP Chemiluminescent Enzyme Immunoassay (CLEIA) exhibited the convenience and good analytical performance necessary for GFAP measurements to aid in the differential diagnosis of neurological disorders. Furthermore, LUMIPULSE G1200 can perform 120 tests/h, suggesting that this system is suitable for high-throughput screening. Additionally, the LUMIPULSE platform has a product portfolio for AD-related biomarkers, allowing sequential measurement of several biomarkers [14,15,16,17]. This GFAP assay would therefore be useful in hospitals and clinical laboratories.

## Figures and Tables

**Figure 1 diagnostics-14-02520-f001:**
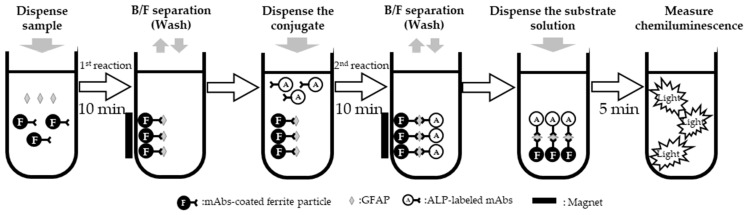
Schematic representation of GFAP measurement using a two-step sandwich method on the LUMIPULSE.

**Figure 2 diagnostics-14-02520-f002:**
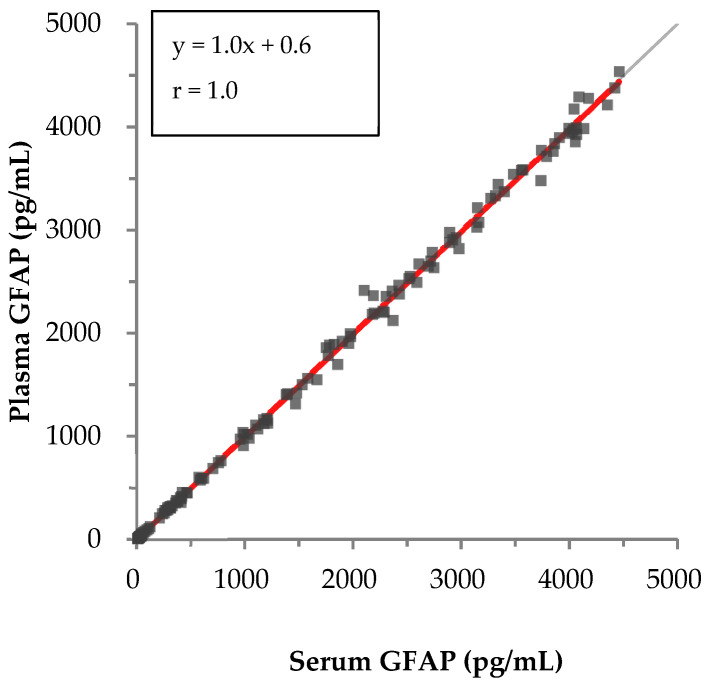
Passing–Bablok regression analysis of serum and plasma correlation. The Passing–Bablok regression method was used to assess the relationship between serum and plasma measurement values of GFAP. The gray line represents the line of identity (y = x), indicating perfect agreement between the two methods. The red line indicates the Passing–Bablok regression line. Equations for the regression line and the correlation coefficient (r) are provided in the figure.

**Figure 3 diagnostics-14-02520-f003:**
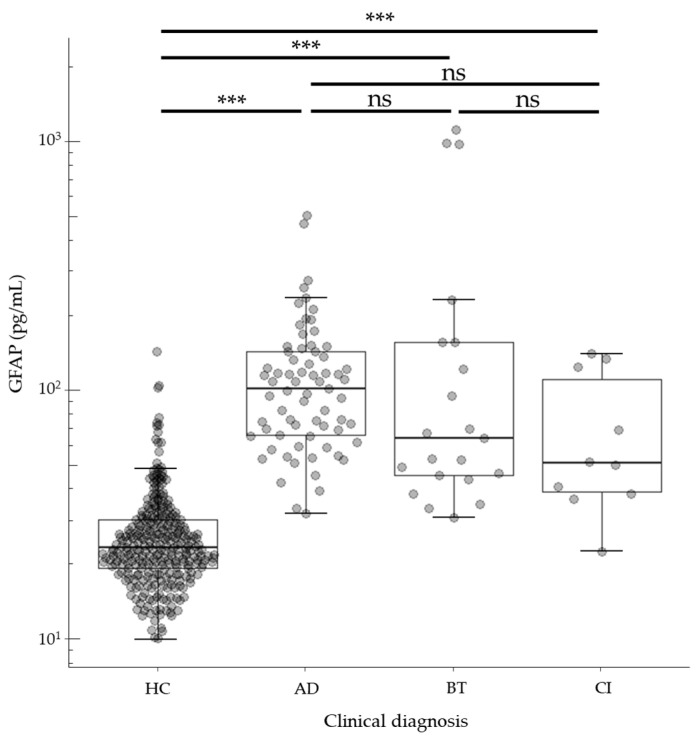
Distribution of GFAP concentration across different disease groups. The figure presents a beeswarm boxplot of GFAP levels. The y-axis is on a logarithmic scale. The boxplot displays median values with the interquartile range (IQR) represented by the lower and upper hinges and whiskers extending to ± 1.5 times the IQR from the first and third quartiles. Data were analyzed using Wilcoxon signed-rank tests with Bonferroni correction. ns: not significant; *** *p* < 0.001.

**Figure 4 diagnostics-14-02520-f004:**
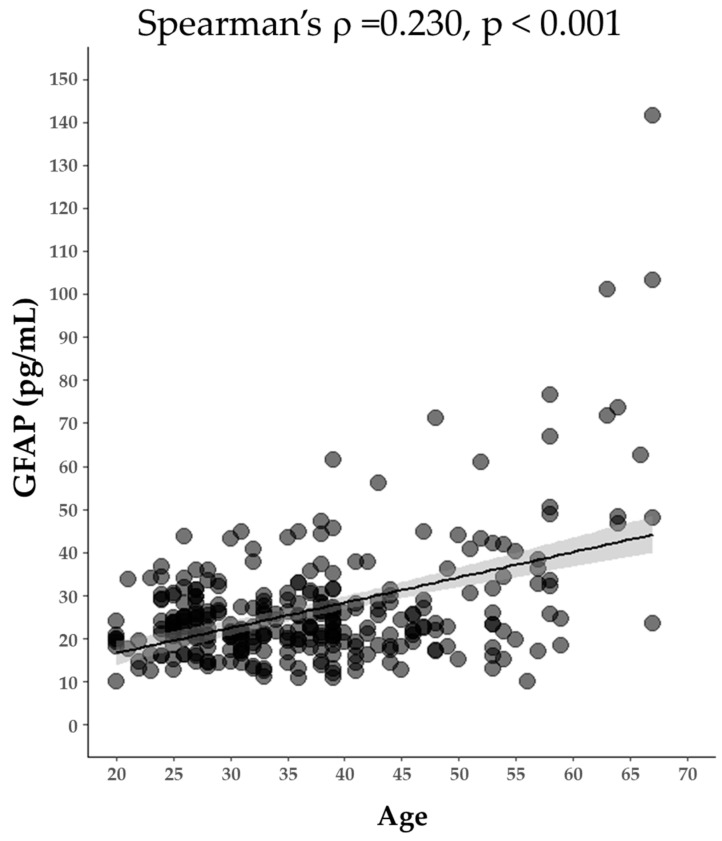
Correlation of age and GFAP concentration in HC group with superimposed linear regression lines (with 95% CI). Statistical methods of Spearman’s rank correlation coefficient were used. *p*-value is indicated; linear regression lines are black.

**Table 1 diagnostics-14-02520-t001:** Features of the measurement system.

	GFAP
Sample type	Serum, Plasma
Sample volume	100 µL
Reportable range	4.0–5000.0 pg/mL
LOD	1.8 pg/mL
LOQ (at 10% CV)	6.0 pg/mL
Calibration points, configuration	2 (master curve), lyophilized

**Table 2 diagnostics-14-02520-t002:** Precision of the GFAP assay.

GFAP Level of Samples	Mean (pg/mL)	Repeatability	Between Run	Between Day	Within Laboratory
Low	108.2	2.5%	1.4%	2.3%	3.7%
Middle	821.5	2.2%	1.7%	3.4%	4.4%
High	3744.4	1.8%	1.6%	2.6%	3.5%

**Table 3 diagnostics-14-02520-t003:** Dilution linearity analysis of the GFAP assay.

	Sample	R-Squared	Slope	Intercept
Serum	1	1.00	1.00 (0.99–1.00)	13.44 (−0.18–27.07)
	2	1.00	1.00 (0.98–1.01)	30.47 (−7.81–68.75)
	3	1.00	1.00 (0.97–1.02)	−0.19 (−71.7–71.31)
	4	1.00	1.00 (0.98–1.01)	13.6 (−25.4–52.59)
	5	1.00	1.00 (0.99–1.00)	17.55 (1.20–33.91)
Plasma	1	1.00	1.00 (0.96–1.04)	39.66 (−60.39–139.71)
	2	1.00	1.00 (0.96–1.03)	49.30 (−43.96–142.57)
	3	1.00	0.99 (0.94–1.04)	71.05 (−60.98–203.07)
	4	1.00	1.00 (0.97–1.02)	34.87 (−24.38–94.12)
	5	1.00	1.00 (0.99–1.00)	7.26 (−8.26–22.77)

**Table 4 diagnostics-14-02520-t004:** Evaluation of interfering substances.

Interfering Substances	Concentration	% Difference from Control		
		GFAP Level of Samples		
		Low	Middle	High
Free Bilirubin	20 mg/dL	−2.7	−1.9	−0.9
Conjugated Bilirubin	20 mg/dL	−0.7	−1.1	0.3
Chyle	1580 FTU	−1.3	0.1	1.3
Hemoglobin	520 mg/dL	−0.5	−0.8	−1.0
Triglycerides	2000 mg/dL	−0.9	−0.1	2.4
Biotin	4250 ng/mL	−0.1	2.1	−0.5

**Table 5 diagnostics-14-02520-t005:** Evaluation of cross reactions.

Cross Reactant	Concentration	Cross-Reactivity Rate (%)	
		Serum	Plasma
rhDesmin	1007 pg/mL	0.1	−0.2
rhNfL	1003 pg/mL	−0.3	−0.3
rhPeripherin	1006 pg/mL	0.2	−0.2
rhVimentin	1009 pg/mL	0.0	−0.6

**Table 6 diagnostics-14-02520-t006:** Characteristics of all specimens by clinical status.

	HC	AD	BT	CI
*n*	296	69	21	10
Sex = Male (%)	121 (40.9)	30 (43.5)	11 (52.4)	7 (70.0)
Age (median [IQR]) †	36.0 [28.0, 43.0]	83.0 [75.0, 88.0]	53.0 [48.0, 68.0]	77.0 [69.0, 79.5]
GFAP (median [IQR]) †	23.1 [19.1, 30.0]	100.5 [65.1, 140.6]	63.4 [45.1, 153.9]	50.4 [38.6, 109.4]

† Median [Q1, Q3].

## Data Availability

The original contributions presented in the study are included in the article, further inquiries can be directed to the corresponding author.

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
