# Peer review of "Glial Fibrillary Acidic Protein’s Usefulness as an Astrocyte Biomarker Using the Fully Automated LUMIPULSE® System"

_diagnostics, 2024, doi:10.3390/diagnostics14222520_

Round 1
Reviewer 1 Report
Comments and Suggestions for Authors
Thank you for giving me this opportunity to review this manuscript. The novelty is clear, and the methodology is sound. The discussion is detailed, and the conclusion is on point.
Revision: Authors should add a sentence in the methods section of the abstract on groups of patients that were used in the study, i.e., healthy control patients and patients with neurodegenerative diseases.
Author Response
Comments1:Authors should add a sentence in the methods section of the abstract on groups of patients that were used in the study, i.e., healthy control patients and patients with neurodegenerative diseases.
Response1: Thank you for your comment. I agree with your comment. We have added the following sentence to the methods section of the abstract: "GFAP levels were measured in 396 serum or plasma samples, comprising both healthy controls and patients with neurodegenerative diseases."
The following typos have been corrected: the email address was incorrect and has been fixed. Additionally, 'r' in line 276 was garbled, and I have corrected it.
Reviewer 2 Report
Comments and Suggestions for Authors
Dear colleagues, You did a great job with GFAP level analysis of in some human disease. The creating of simple, standart - based express analytics kit can only be welcomed, especially in the conditions of a large multidisciplinare hospital. It's especially important that reference values have been obtained for various neurological and neurosurgical pathologies. The differential diagnosis of Alzheimer's disease, sclerosis disseminatus and more acute brain injuries receives an express assessment. Of course, a large volume of clinical material is necessary, but the date obtained are statistically significant.
Author Response
Comments: Dear colleagues, You did a great job with GFAP level analysis of in some human disease. The creating of simple, standart - based express analytics kit can only be welcomed, especially in the conditions of a large multidisciplinare hospital. It's especially important that reference values have been obtained for various neurological and neurosurgical pathologies. The differential diagnosis of Alzheimer's disease, sclerosis disseminatus and more acute brain injuries receives an express assessment. Of course, a large volume of clinical material is necessary, but the date obtained are statistically significant.
Response: Thank you for your thoughtful feedback on our work regarding GFAP level analysis in human diseases. We appreciate your recognition of the importance of creating a standardized diagnostic kit for rapid assessment, especially in a multidisciplinary hospital setting. We agree that obtaining reference values for various neurological and neurosurgical pathologies is crucial for accurate differential diagnosis, particularly for conditions such as Alzheimer’s disease and multiple sclerosis.
We have added the following sentence to the methods section of the limitations line 296-296, “Future studies should investigate patient specimens across a range of medical conditions within a multidisciplinary hospital setting to establish cutoff points.”
Thank you once again for your valuable insights.
This is a minor change: the terminology 'measurement range' in Table 1 has been revised to 'reportable range' to reflect the assay's output range.
Reviewer 3 Report
Comments and Suggestions for Authors
The article is focused on a new technique for evaluating GFAP in blood. The technique is well described, all necessary procedures are given to evaluate the possibility of using this method, specificity, precision, cross reactivity are evaluated.
The manuscript could be improved by adding the following information. The authors should add more about the role of GFAP in the pathogenesis of neuropsychiatric diseases and whether the determination of GFAP is used in the clinic. For example, the authors note that the GFAP assay has received FDA approval for TBI. It should also be written whether this assay is available in the clinical guidelines of other countries. Can GFAP be determined in other biological samples, such as cerebrospinal fluid? Write more prospects for the use of your assay, its cost in comparison with other methods of detection.
Author Response
Response1: Thank you for your comment. I agree with your comment. As for cost, I believe it will be comparable to that of Abbott's GFAP, which is not a significant advantage. On the other hand, I can describe the availability of FDA-approved tests, which we have added the following sentences.
Line 75: "Furthermore, it is not approved as an IVD tool in Japan or the EU, and not widely distributed at least in Japan."
Line299-301: “Fifth, although we reported the measurement of serum and plasma samples in this study, we acknowledge that the measurement range for CSF samples needs to be expanded, which will be addressed in future research.”
The following typos have been corrected: the email address was incorrect and has been fixed. Additionally, 'r' in line 276 was garbled, and I have corrected it.